# Abnormal Gait Pattern Examination Screening for Physical Activity Level after One Year in Patients with Knee Osteoarthritis

**DOI:** 10.3390/jfmk8010024

**Published:** 2023-02-15

**Authors:** Shunsuke Yamashina, Kazuhiro Harada, Ryo Tanaka, Yu Inoue

**Affiliations:** 1Department of Rehabilitation, Taira Hospital, Okayama 709-0498, Japan; 2Graduate School of Humanities and Social Sciences, Hiroshima University, Hiroshima 739-0046, Japan; 3Graduate School of Health Sciences, Kibi International University, Okayama 716-0018, Japan

**Keywords:** knee osteoarthritis, abnormal gait pattern, physical activity, predictive validity

## Abstract

This study examined the relationship between abnormal gait pattern and physical activity level one year later in patients with knee osteoarthritis (KOA) and determined the clinical utility of the abnormal gait pattern examination. Initially, the patients’ abnormal gait pattern was assessed using seven items, based on the scoring system reported in a previous study. The grading was based on a three-criteria system with 0: no abnormality, 1: moderately abnormal, and 2: severely abnormal. The patients were then classified into three groups according to physical activity level one year after gait pattern examination: low, intermediate, and high physical activity groups, respectively. Cut-off values for physical activity levels were calculated based on abnormal gait pattern examinations results. On follow-ups with 24 of the 46 subjects, age, abnormal gait pattern, and gait speed showed significant differences among the three groups according to the amount of physical activity. Effect size of abnormal gait pattern was higher than age and gait speed. Patients with KOA with physical activity < 2700 steps/day and <4400 steps/day at one year had abnormal gait pattern examination scores of ≥8 and ≥5, respectively. Abnormal gait pattern is associated with future physical activity. The results suggested that abnormal gait pattern examinations in patients with KOA could be used to screen for the possibility of physical activity being <4400 steps one year later.

## 1. Introduction

Patients with knee osteoarthritis (KOA) tend to be less physically active than healthy individuals [1]. Low physical activity leads to muscle weakness and impaired joint function. Previous reports have shown that patients with KOA took approximately 2300 steps/day less compared to healthy subjects [2], especially those with pain [3]. It is important for patients with KOA to include mild to moderate physical activity, such as gait and full-body exercise, to prevent the worsening of their disease severity and ability to perform daily activities [4]. Thus, prevention of decreased physical activity may lead to an increase in daily living abilities of patients with KOA and reduced disease severity.

Early detection of functional and structural impairment of KOA is considered important to prevent early reduction in physical activity. Impairment due to KOA is conventionally assessed using radiographic imaging as well as muscle strength and joint range of motion measurements. These examinations may detect impairments, such as joint deformity, muscle weakness, and limited joint range of motion. However, these impairments cannot be accurately detected until the severity of KOA has progressed to a certain degree [5]. Therefore, conventional examinations are not useful in the early prevention of reduced physical activity due to KOA.

Additionally, impairment due to KOA includes abnormal gait pattern. Gait pattern is a motor function defined by the International Classification of Functioning, Disability and Health (ICF) [6]. The findings of abnormal gait patterns in KOA include lateral knee thrust during the stance phase of gait [7], lateral trunk shift [8], and increased ankle eversion angle [9]. These findings are observed from the early stages [10] and are associated with pain and severity [11,12]. Based on these findings, examinations that detect abnormal gait patterns can identify impairments due to KOA at an earlier stage than conventional examinations. Therefore, since impairment is related to the amount of physical activity, abnormal gait pattern examination could be useful in preventing decrease in physical activity in patients with KOA at an early stage. Although pain and impaired lower limb function (mainly muscle strength) have been reported as factors that decreased the amount of physical activity [13], to our knowledge, no studies have been conducted using abnormal gait patterns to examine this decrease.

The aim of this study was to examine the relationship between abnormal gait patterns and physical activity in KOA patients and investigate the clinical usefulness of abnormal gait pattern examination. The hypothesis of our study was that abnormal gait patterns were related to future decrease in physical activity.

## 2. Materials and Methods

### 2.1. Subjects

The subjects were men and women aged 60 years or older with medial-type KOA on conservative treatment from medical institutions. The inclusion criterion was defined as being able to gait indoors alone for more than 10 m. Patients with a cognitive decrease (Hasegawa Dementia Rating Scale-Revised score less than 20 points [14]) and a history of central nervous system disease (which resulted in motor paralysis) were excluded from the study.

### 2.2. Measurements and Methods

Age and sex were investigated as basic attributes. The Kellgren–Lawrence classification (K-L classification) [15] was investigated as medical information. Physical function was examined regarding knee joint flexion range of motion, knee joint extension and flexion muscle strength, abnormal gait pattern, gait speed, and daily physical activity.

The knee joint range of motion was measured using a goniometer. Knee joint muscle strength was measured in newton (N) using a hand-held dynamometer and adjusted for leg length. Body weight ratios were then calculated (Nm/kg). Video data recorded by a video camera were used to examine abnormal gait patterns. Two cameras were used to examine movement in the sagittal and frontal planes. Physical activity was measured regarding the number of steps/day taken via an activity meter (Active Style Pro HJA-350 IT, OMRON Inc. Okayama, Japan). The activity meter was worn on the waist throughout the day except during sleeping, bathing, and exercising with heavy contact, and the measurements were taken for seven days. Worsening of physical activity due to KOA after two years is considered gradual yet exacerbated by pain and joint symptoms [16]. In this study, the follow-up period was set shorter than in previous studies, at one year. The reason for this was to ascertain whether a decrease in activity level occurred early on and to identify any abnormal gait patterns related to this decrease.

#### Abnormal Gait Pattern Examination

The measurements used for the examination were the following seven items: Increased foot progression angle in the early stance phase, decreased foot contact angle in the early stance phase, decreased ankle plantar flexion in the late stance phase, increased knee adduction in the early stance phase, decreased knee extension in the mid-stance phase, decreased knee flexion in the swing phase, and decreased hip extension in the late stance phase (Table A1) [17,18].

The grading was based on a three-criteria system with 0: no abnormality, 1: moderately abnormal, and 2: severely abnormal. The highest and lowest scores were 14 and 0, respectively. The higher the score, the more severe the abnormality in the gait pattern. Measurements were taken on the frontal plane and sagittal plane with a video camera. The distance of the gait path was 7.6 m, with 1 m to spare at each end. The video camera captured images at a distance of 3 m in the frontal plane and 5 m in the sagittal plane, where the whole body was visible. Furthermore, the focal point was set at 1.2 m from the floor. In principle, gait support devices were not used. The abnormal gait pattern was graded using the images taken. They were replayed and graded multiple times. In the case of bilateral KOA, the lower extremity with stronger clinical signs (judged by K-L classification, muscle strength, and pain) was defined as the disabled side. The raters were two persons, engaged in musculoskeletal physical therapy, not involved in data analysis. The two raters jointly evaluated scoring.

### 2.3. Statistical Analysis

Differences in descriptive statistics between subjects who could be followed-up and those who could not were analyzed with an uncorrelated *t*-test.

The predictive validity of the abnormal gait pattern examination was based on the association between the total score of abnormal gait pattern examination at baseline and amount of physical activity one year later. After one year, physical activity was classified into three groups: low activity group (<2700 steps/day), intermediate activity group (2700–4399 steps/day), and high activity group (>4400 steps/day) [19].

The relationship between the amount of physical activity and age, knee joint flexion range of motion, knee joint extension and flexion muscle strength, gait speed, and abnormal gait pattern was examined. A one-way analysis of variance was performed, followed by multiple comparison tests. Receiver operator characteristic (ROC) analysis was also performed to calculate the cut-off value of the abnormal gait pattern examination for decreased physical activity. The analyses were conducted with the event occurrence defined as less than 2700 steps/day and 4400 steps/day of physical activity after one year, respectively. The Youden Index was used as the calculation method. SPSS statistics version 24 (IBM Japan, Tokyo, Japan) was used for statistical analysis.

#### Sample Size Estimates

In a previous study, when physical activity daily was classified by quartile, the second quartile (50% of the subjects) was 4363 steps/day and the first quartile (25% of the subjects) was 2718 steps/day [19]. Based on this report, it was assumed that the ratio of event occurrence (physical activity less than 4400 steps/day after one year) to no event occurrence (physical activity greater than 4400 steps/day after one year) was 1 (50%) to 1 (50%). The ratio of the event occurrence for less than 2700 steps/day was assumed to be 1 (25%) to 3 (75%). In all cases, the significance level was set at 0.05, area under the curve (AUC) was set at 0.8, and power was set at 0.8 to estimate the sample size. The results of the estimation showed that for the model with less than 4400 steps/day, a total of 20 cases were required, 10 each with and without events. In the model with less than 2700 steps/day, a total of 27 cases were required, 7 with event occurrence and 20 without event occurrence. Statistical analysis software R version 4.0.2 (CRAN, freeware) pROC package was used.

## 3. Results

### 3.1. Subject Descriptive Statistics

In total, 46 patients with KOA met the inclusion criteria and cooperated. Of these, 24 patients available for follow-up at one year were included in the analysis. In the model with less than 2700 steps/day, 27 cases were calculated as the required sample size. However, data collection was difficult during the study period; hence, the measurements were discontinued. Other main characteristics are summarized in Table 1.

### 3.2. Relationship of Physical Activity

An analysis of variance showed that age, abnormal gait pattern, and gait speed were significantly different among the three groups. Multiple comparisons found significant differences in age between the low and intermediate physical activity groups and low and high physical activity groups. Regarding abnormal gait patterns, significant differences were found between the low and high physical activity groups and intermediate and high physical activity groups. Additionally, gait speed was significantly different between the low and high physical activity groups (Table 2). In particular, abnormal gait patterns showed a high effect size with an F value of 14.62.

### 3.3. Predictive Validity of Gait Pattern

According to the results of the ROC analysis, the cut-off value of the abnormal gait pattern examination was 8 points when event occurrence was defined as less than 2700 steps/day of physical activity after one year. The ROC area under the curve (AUROC) was 0.91, sensitivity 1.00, and specificity 0.89. The positive likelihood ratio (PLR) was 9.09, and the negative likelihood ratio (NLR) was 0. In contrast, the cut-off value of the abnormal gait pattern examination was 5 points when event occurrence was defined as less than 4400 steps/day of physical activity after one year. The AUROC was 0.89, sensitivity 0.92, and specificity 0.91. The PLR was 10.2 and the NLR was 0.09 (Figure 1, Table 3). A contingency table was created with reference to the cut-off values of each model, and the posterior odds were calculated. For the models in which the event occurred at less than 2700 steps/day and 4400 steps/day, the positive odds were 3.0 and 11.82 and the negative odds were 0 and 0.11, respectively (Table 4).

## 4. Discussion

In this study, the relationship between abnormal gait patterns and physical activity in KOA was investigated to determine the clinical utility of the abnormal gait pattern examination. The study hypothesis was that abnormal gait patterns would be associated with lower future physical activity. After one year, the patients were classified into three groups: low group (less than 2700 steps/day), intermediate group (2700–4399 steps/day), and high group (4400 steps/day or more), and the relationship between physical activity and physical function was examined. The physical functions that showed significant differences from the amount of physical activity were age, abnormal gait pattern, and gait speed. Among these factors, the effect size of abnormal gait pattern was higher compared to that of other factors. Based on the results of the abnormal gait pattern examination, a cut-off value was calculated for KOA patients prone to low physical activity after one year.

The validity of this study is described. Twenty-four of the 46 subjects were available for follow-up from baseline. Compared to the subjects who were difficult to follow up, no differences were found in physical characteristics, such as muscle strength and range of motion, and age. Thus, it can be concluded that there was no bias in the physical characteristics of the sample that were available for follow-up. The data of physical activity in this study were 4803 steps/day at baseline and 4363 steps/day at one year. In a previous study, the amount of physical activity for KOA was estimated to be approximately 4400 steps/day [2] and was considered similar to our data. Based on these facts, the follow-up and physical activity data were obtained from an unbiased sample. In previous studies, detecting abnormal gait patterns (knee adduction) has been shown to be predictive of progression after 18 months [12]. In this study, abnormal gait patterns were also found to predict a decrease in physical activity at 12 months. These results suggest that abnormal gait patterns are a predictor of physical function in the future.

The novelty of this study lies in the identification of the possibility that abnormal gait patterns may affect the amount of future physical activity. Impairment due to KOA has traditionally been assessed using radiographic imaging, muscle strength, and joint range of motion. These examinations could not accurately detect these impairments until the severity progressed to a certain degree [5]. Although pain and decreased lower limb function were reported as factors that decreased the amount of physical activity [13], no reports that used abnormal gait patterns to examine this decrease were identified. The results of this study showed that age, abnormal gait pattern, and gait speed were significantly associated with decreased physical activity. Furthermore, the effect size of abnormal gait pattern was higher. Therefore, abnormal gait pattern was a factor that influenced a decrease in physical activity and could be useful during screening to predict the prognosis.

Some results can be used to predict the amount of physical activity one year later based on the scores of the abnormal gait pattern examination. The ROC analysis showed that patients with KOA with physical activity less than 2700 after one year had abnormal gait pattern examination scores of at least 8 points. The PLR was 9.09 and the NLR was 0 with a sensitivity of 1.00 and specificity of 0.89. The posterior odds were positive odds 3.0 and negative odds 0. Using these posterior odds as probabilities, there was a 75% probability that a patient would take less than 2700 steps/day one year later if they scored 8 or more points on the abnormal gait pattern examination. The results also suggested that the probability of physical activity being lesser than 2700 steps/day was 0% when the score was lesser than 8 points. Similarly, patients with KOA with physical activity less than 4400 after one year had abnormal gait pattern examination scores of at least 5 points. The PLR was 10.2 and the NLR was 0.09 with a sensitivity of 0.92 and specificity of 0.91. The posterior odds were positive odds 12.03 and negative odds 0.11. Using these posterior odds as probabilities, there was a 92% probability that a patient would take less than 4400 steps/day one year later if they scored 5 or more points on the abnormal gait pattern examination. The results also suggested that the probability of physical activity being less than 4400 steps/day was 10% when the score was lesser than 5 points. The above results suggest that the model has predictive validity for determining subjects likely to have low physical activity levels.

There are two limitations in this study. First, the sample size was small. In conducting the ROC analysis, a sample size estimate was made, and the required sample size for the model in which physical activity less than 2700 was considered an event occurrence was 27. The sample was, therefore, not sufficiently validated for the possibility of true positives and false negatives. Second, in this study, the follow-up period was shorter than the follow-up periods reported in previous studies [16]. Therefore, the physical characteristics of patients with KOA, whose physical activity decrease after one year, were unclear. However, despite these limitations, the examination characteristics of the model with an event occurrence of less than 4400 steps/day was good, and preventive measures can be considered for the decrease in physical activity after one year.

## 5. Conclusions

The aim of this study was to examine the relationship between abnormal gait patterns and amount of physical activity with KOA and determine the clinical utility of abnormal gait patterns examination. Results showed that abnormal gait pattern was a factor related to future physical activity. The results of this study suggested that abnormal gait pattern examinations in patients with KOA could be used to screen for the possibility of physical activity being less than 4400 steps one year later. These findings suggested that abnormal gait patterns examination in patients with KOA had predictive validity for identifying subjects likely to have less physical activity in the future.

## Figures and Tables

**Figure 1 jfmk-08-00024-f001:**
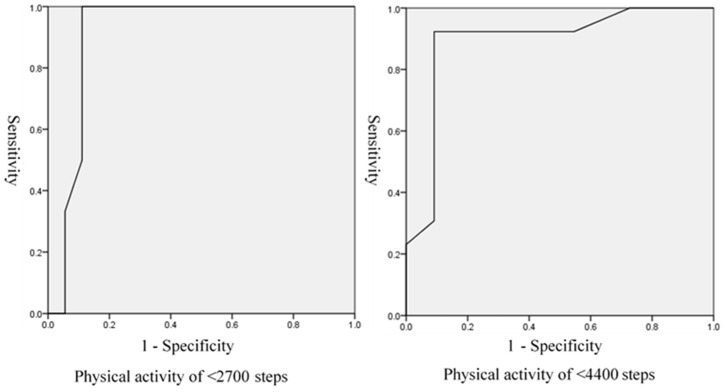
ROC analysis of gait pattern examination.

**Table 1 jfmk-08-00024-t001:** Subject descriptive statistics.

	Follow-Up Group n: 24	Dropout Group n: 22	*p*-Value
Age, years	76.33 (8.02)	70.36 (6.19)	0.01
Sex	Men: 5, Women: 19	Men: 5, Women: 17	N/A
BMI	23.42 (3.97)	27.11 (5.51)	0.012
K-L severity	I: 10, II: 6, III: 3, IV: 5	I: 4, II: 9, III: 6, IV: 3	N/A
Knee flexion ROM, °	131.88 (12.23)	132.73 (6.70)	0.78
Knee extension muscle strength, Nm/kg	0.79 (0.39)	0.78 (0.37)	0.94
Knee flexion muscle strength, Nm/kg	0.43 (0.17)	0.42 (0.23)	0.92
Abnormal gait pattern	5.46 (3.16)	4.27 (2.60)	0.18
Gait speed, m/s	1.22 (0.36)	1.24 (0.20)	0.76
Physical activity, steps/day			
Baseline	4803.50 (2785.80)	3862.0 (1925.0)	0.19
After 1 year	4363.21 (2779.80)		N/A

Mean (SD). BMI: body mass index, K-L: Kellgren–Lawrence, ROM: range of motion, N/A: Not Applicable, SD: standard deviation.

**Table 2 jfmk-08-00024-t002:** Relationship of physical activity.

	Physical Activity			F-Value	*p*-Value
	Low Group n: 6	Intermediate Group n: 7	High Group n: 11		
Age, years	84.83 (5.74)	74.42 (7.81)	72.91 (6.51)	6.46	0.01 * †
Knee flexion ROM, °	128.33 (9.83)	137.14 (8.09)	130.45 (15.56)	0.93	0.41
Knee extension muscle strength, Nm/kg	0.57 (0.16)	0.84 (0.51)	0.88 (0.39)	1.35	0.28
Knee flexion muscle strength, Nm/kg	0.35 (0.18)	0.40 (0.10)	0.49 (0.19)	1.58	0.23
Abnormal gait pattern	12.67 (1.28)	9.00 (1.19)	4.27 (0.95)	14.62	0.00 † ‡
Gait speed, m/s	0.83 (0.17)	1.17 (0.31)	1.46 (0.28)	10.92	0.001 †

Mean (SD). Post-hoc comparisons, *: low group vs intermediate group < 0.05, †: low group vs high group < 0.05, ‡: intermediate group vs high group < 0.05; low group: <2700 steps/day, intermediate group: 2700–4399 steps/day, high group: ≥4400 steps/day; ROM: range of motion, SD: standard deviation, vs: versus.

**Table 3 jfmk-08-00024-t003:** Test characteristics of the gait pattern.

	AUC	Cut-Off Value	Sensitivity	Specificity	PLR	NLR
<2700 steps	0.91	8	1.00	0.89	9.09	0
<4400 steps	0.89	5	0.92	0.91	10.2	0.09

AUC: area under the curve, PLR: positive likelihood ratio; NLR: negative likelihood ratio.

**Table 4 jfmk-08-00024-t004:** Contingency table of the gait pattern.

	Decrease in PA <2700 Steps	No Decrease in PA ≥2700 Steps	Decrease in PA <4400 Steps	No Decrease in PA ≥4400 Steps
Test positive	6	2	12	1
Test negative	0	16	1	10

Model with <2700 steps: prior odds (6 + 0)/(2 + 16) = 0.33, positive posterior odds 0.33 × 9.09 (PLR) = 3.0, negative posterior odds 0.33 × 0 (NLR) = 0. Model with <4400 steps: prior odds (12 + 1)/(1 + 10) = 1.18, positive posterior odds 1.18 × 10.2 (PLR) = 12.03, negative posterior odds 1.18 × 0.09 (NLR) = 0.11. PLR: positive likelihood ratio, NLR: negative likelihood ratio, PA: physical activity.

## Data Availability

MDPI Research Data Policies at https://www.mdpi.com/ethics (accessed on 13 February 2023).

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
