# Peer review of "Abnormal Gait Pattern Examination Screening for Physical Activity Level after One Year in Patients with Knee Osteoarthritis"

_jfmk, 2023, doi:10.3390/jfmk8010024_

Round 1

Reviewer 1 Report

This study examined the relationship between abnormal gait patterns and physical activity in patients with knee osteoarthritis. The topic is very interesting, and the manuscript is well organized. However, language should be polished by native English speakers to make it more transparent and understandable.

INTRODUCTION

L52-53: It is necessary to include a reference to support this idea.

L52-53: The authors stated that no studies have been conducted using abnormal gait  patterns to examine this decrease. This statement should be made with caution.

METHOD

L67-68The inclusion criteria is very limited. It should be more specific and contains more conditions.

L74-79Descriptive statistics should be moved to the Statistical Analysis section.

L97-109: Whether Abnormal gait pattern score was rated by the two raters together or the mean score of the two raters

RESULT

L180: The methods section did not describe collecting the Knee extension/ flexion muscle strength data.

L194 There are some typographical errors in Table 4.

DISCUSSION

The author did not compare the data with other studies in more detail.

Reviewer 2 Report

The paper presents an interesting research study that examined the relationship between abnormal gait pattern and physical activity in patients with knee osteoarthritis (KOA) and determined the clinical utility of abnormal gait pattern examination. Several changes are required to recommend this submission for publication in “JFMK” given that contents in the manuscript are wordy and hard to read. Moderate editing of the manuscript and concise writing are required. Specifically, the title, abstract, and introduction of the study require editing in the English language. The syntax and choice of words are not quite right. . Some examples are highlighted in the specific comments below, but the whole paper should be checked.

Abstract

Lines 13-14: “ according to activity level after one year…” Do you mean one year after baseline measurements?

Lines 14-15: “The abnormal gait pattern was assessed by using the scoring system reported in a previous study” This should be described and be more clear to the reader.

 Lines 15-18: “Cut-off values for KOA for low activity after one year from baseline were calculated based on the results of abnormal gait pattern examinations. On follow-ups with 24 of the 46 subjects, age, abnormal gait pattern, and gait speed showed significant differences among age groups”. “Cut-off values for KOA for low activity” should be revised. “Age ….showed significant differences among age groups” does not make sense. Please modify it as well.

Lines 18-19: “Effect size of abnormal gait pattern was higher than other factors.” Which factors are you referring to? Be precise.

 Lines 19-20: “Patients with KOA with physical activity <2700 and <4400…” Add steps/day after 2700 and 4400, respectively.

Introduction

Line 27: “physical activity” should be replaced with “physical or physically active”.

Line 30: “lesser” should be “less”.

Lines 30-32: “In addition, patients with KOA avoid physical activity due to pain, causing muscle weakness and joint instability, leading to activity limitations [1]”. This sentence has almost identical meaning to the previous one. Please revise.

Lines 39-40: “assessed using radiographic imaging, muscle strength, and joint range of motion.” Please revise. For example: “assessed using radiographic imaging as well as muscle strength and joint range of motion measurements.”

Line 45: “In contrast, impairment due to KOA includes abnormal gait pattern.” Why “in contrast”? Please replace it. For example: “Additionally, impairment due to KOA…”

Line 47: “Suspicious findings” Replace suspicious with a more suitable word.

Lines 58-59: “The hypothesis of our study was that abnormal gait patterns were related to future decrease in physical activity”. The hypothesis should be moved after the aim of the study.

Lines 59-61: “The aim of this study was to examine the relationship between abnormal gait patterns and physical activity in KOA and investigate the clinical usefulness of abnormal gait patterns examination.” Please add the word “patients” after the word “KOA”.

Lines 68-71: “Exclusion criteria  were those with cognitive decrease (Hasegawa dementia rating scale-revised score less than 20points [14]) and a history of central nervous system disease (which resulted in motor paralysis).” This sentence should be revised. For example, "Patients with a cognitive decrease (Hasegawa dementia rating scale-revised score less than 20 points [14]) and/or a history of central nervous system disease (which resulted in motor paralysis) were excluded from the study”.

Line 75: “…was investigated as medical attributes”. “…as medical attribute”? The word attribute could be replaced with a more appropriate word.

Line 80: “Video data recorded by a video camera were used to examine abnormal gait patterns.” How many cameras did you use to examine frontal and sagittal plane motion?

Line 84: “Worsening of activity limitations”. Please delete the word “limitations” and add the word “physical” before the word “activity”.

Lines 102-103: “where the sole of the foot could be seen to the top of the head.” What do you mean in this sentence?

Lines 104-106: “The abnormal gait pattern was graded using the images taken once, and they were replayed at least twice. There was no upper limit to the number of observations”. I am not sure I clearly understand the meaning of this phrase. Please revise.

Results

Line 168: “The PLR was 10.2 and the NLR was 0.09. (Figure1, Table 3) A contingency…” Delete the full stop after “0.09” and add it following “(Figure1, Table 3)”.

Discussion

Line 218: “…was calculated for KOA for individuals…” Please revise.

Round 2

Reviewer 1 Report

I agree to accept the current version for publication.

Author Response

The document was revised using a native editorial proofreading service.

Reviewer 2 Report

The paper is improved. Please revise the following.  

Title:

I suggest the title should be changed to “Abnormal gait pattern examination screening for physical activity level after one year (or one year later) in patients with knee osteoarthritis”

Abstract:

Line 12: Please add “level one year later” before “in patients with knee osteoarthritis (KOA)”.

Line 13: “All the subjects had KOA” is unnecessary. You have already stated this in line 12. Please change to “The patients were classified into three groups according to activity level…”    

Line 15-16: I suggest you change the following sentences “The abnormal gait pattern was assessed by using the scoring system reported in a previous studyThis scoring system were seven items” to “The abnormal gait pattern was assessed using seven items, based on the scoring system reported in a previous study”.  

Line 20: Please change “among three groups” to “among the three groups”.

The abstract is hard to read. I suggest being revised as follows:

This study examined the relationship between abnormal gait pattern and physical activity level one year later in patients with knee osteoarthritis (KOA) and determined the clinical utility of the abnormal gait pattern examination. Initially, the patients’ abnormal gait pattern was assessed using seven items, based on the scoring system reported in a previous study. The grading was based on a three-criteria system with 0: no abnormal, 1: moderately abnormal, and 2: severely abnormal. The patients were then classified into three groups according to physical activity level one year after gait pattern examination: low, intermediate, and high physical activity groups, respectively. Cut-off values for physical activity levels were calculated based on abnormal gait pattern examination results. On follow-ups with 24 of the 46 subjects, age, abnormal gait pattern, and gait speed showed significant differences among the three groups according to the amount of physical activity.” The rest lines of the abstract are ok.

Line 112: Revise “whole body visible” to “ whole body was visible”.

Lines 230-231: Please change “a cut-off value was calculated for of KOA for those individuals prone…” to “a cut-off value was calculated for KOA patients prone to low physical activity after one year”.   

Author Response

(The authors gave the same response as above.)
